# Listening Effort in Tinnitus: A Pilot Study Employing a Light EEG Headset and Skin Conductance Assessment during the Listening to a Continuous Speech Stimulus under Different SNR Conditions

**DOI:** 10.3390/brainsci13071084

**Published:** 2023-07-17

**Authors:** Giulia Cartocci, Bianca Maria Serena Inguscio, Giovanna Giliberto, Alessia Vozzi, Andrea Giorgi, Antonio Greco, Fabio Babiloni, Giuseppe Attanasio

**Affiliations:** 1Department of Molecular Medicine, Sapienza University of Rome, 00161 Rome, Italy; giliberto.1795255@studenti.uniroma1.it (G.G.); fabio.babiloni@uniroma1.it (F.B.); 2Department of Research and Development, BrainSigns Ltd., 00198 Rome, Italy; biancams.inguscio@uniroma1.it (B.M.S.I.); alessia.vozzi@uniroma1.it (A.V.); andrea.giorgi@uniroma1.it (A.G.); 3Department of Human Neuroscience, Sapienza University of Rome, 00185 Rome, Italy; 4SAIMLAL Department, Sapienza University of Rome, 00185 Rome, Italy; 5Department of Sense Organs, Sapienza University of Rome, 00161 Rome, Italy; antonio.greco@uniroma1.it; 6Department of Computer Science, Hangzhou Dianzi University, Hangzhou 310005, China; 7Head and Neck Department, Policlinico Umberto I, 00161 Rome, Italy; giuseppe.attanasio@uniroma1.it

**Keywords:** EEG, skin conductance, tinnitus, alpha values, continuous speech, background noise, hyperacusis

## Abstract

Background noise elicits listening effort. What else is tinnitus if not an endogenous background noise? From such reasoning, we hypothesized the occurrence of increased listening effort in tinnitus patients during listening tasks. Such a hypothesis was tested by investigating some indices of listening effort through electroencephalographic and skin conductance, particularly parietal and frontal alpha and electrodermal activity (EDA). Furthermore, tinnitus distress questionnaires (THI and TQ12-I) were employed. Parietal alpha values were positively correlated to TQ12-I scores, and both were negatively correlated to EDA; Pre-stimulus frontal alpha correlated with the THI score in our pilot study; finally, results showed a general trend of increased frontal alpha activity in the tinnitus group in comparison to the control group. Parietal alpha during the listening to stimuli, positively correlated to the TQ12-I, appears to reflect a higher listening effort in tinnitus patients and the perception of tinnitus symptoms. The negative correlation between both listening effort (parietal alpha) and tinnitus symptoms perception (TQ12-I scores) with EDA levels could be explained by a less responsive sympathetic nervous system to prepare the body to expend increased energy during the “fight or flight” response, due to pauperization of energy from tinnitus perception.

## 1. Introduction

Listening effort is a hot topic in hearing research, especially concerning its implication in hearing-impaired patients, hearing aid, and cochlear implant users. Listening effort has been defined as “The mental exertion [effort] required to attend to, and to understand, an auditory message [listening]” [1] and “The deliberate allocation of mental resources to overcome obstacles in goal pursuit when carrying out a [listening] task.” (cfr Kahneman 1973—Limited capacity model) [2].

Typical measures of listening effort could be divided into (i) measures of the activity of the central nervous system, like electroencephalography (EEG) [3,4], functional magnetic resonance imaging (fMRI) [5], and fNIRS [6,7]; (ii) measures of the activity of the peripheral nervous system, like pupillometry [8,9] and electrodermal activity (EDA) [10,11]; (iii) questionnaires like the Speech, Spatial, and Quality of Hearing Scale (SSQ) [12], the NASA Task Load Index (NASA-TLX) [11] and ratings [13]; and (iv) dual tasks [14].

However, concerning neurophysiological measures, EEG and pupillometry appear as the most employed techniques, recently co-recorded [15,16] and given a complementary but not overlapping sensitivity to listening effort dimensions. Concerning EEG, the most employed index of listening effort is the variation of alpha rhythm [2] (approximately 8–13 Hz), decreasing during the active processing of language stimuli [17] and possibly involved in a “gating by inhibition” mechanism, aimed at inhibiting task-irrelevant activities in task-irrelevant regions [18]. The parietal area appears to present a clear role in listening effort processes, with alpha levels increasing with the difficulty level of the audibility conditions [3,19,20,21,22,23]. The extent of alpha activity decrease is linked to speech intelligibility [22,24], according to the hypothesis of alpha’s anticipatory/preparatory role for the arrival of expected stimuli [25]. However, there is evidence of decreasing alpha activity in correspondence with more difficult hearing conditions [16,26]. Specifically, the areas of interest in such alpha increases are central-parietal [4,27] and occipital-parietal [28]. The alpha rhythm as an index of the listening effort in speech-in-noise recognition tasks employing different signal and noise directions has already been studied in hearing loss patients, for instance, in children with asymmetric hearing loss [20], in single-side deaf children [29], in adult unilateral cochlear implant (UCI) users [21,30], and also in the comparison of different cochlear implant (CI) processors, to identify the tools eliciting the lower listening effort [31,32,33].

Tinnitus, one of the most common otological symptoms [34], is generally defined as an auditory perception in the absence of environmental sound stimulation [35,36]. Tinnitus can be divided into two broad groups, objective and subjective [37]. Objective tinnitus is caused by sound generated in the body reaching the ear through conduction in body tissues [38]. Subjective tinnitus is a perception of a phantom sound heard by a person in the absence of any external physical stimulation [39]. The pathophysiological background and causes of tinnitus are not well-determined [40] and can be categorized into primary and secondary cases [41]. Primary tinnitus is idiopathic and can or cannot be combined with sensorineural hearing loss. The secondary tinnitus forms are indeed associated with an identifiable organic cause. Such causes include cerumen impaction to otosclerosis, cochlear abnormalities, and auditory nerve and central nervous system pathologies [42]. Considering the scientific investigation of EEG correlations in tinnitus patients, to the best of our knowledge, there is only one study [43] concerning EEG assessment of listening effort in tinnitus patients employing the MATRIX test as stimuli and the Tinnitus Handicap Inventory (THI) [44] as a questionnaire for tinnitus symptom-related distress. The present study decided to employ a more ecologic audiobook, given the relevance of using such stimuli, which are more similar to everyday communicative conditions, as measures of listening effort. In addition to EEG, we employed EDA as a measure of skin conductance and the visual analog scale (VAS) scale as a measure of rated difficulty. Moreover, in addition to THI [45], we employed TQ12-I to assess tinnitus-related distress [46]. An EDA measure was included, given the emotional impact of tinnitus on patients and the emotional sensitivity of such a measure of listening effort, which is already employed in several studies on emotions [47,48]. EDA has been employed as a listening effort index, showing increased reactivity in response to degraded auditory conditions compared to non-degraded ones. However, performances and listeners’ perceptions of task demand were comparable across the degraded conditions [49]. Furthermore, skin conductance reactivity to background noise compared to quiet conditions was higher for hearing-impaired than normal hearing persons [10]. Finally, given the frequent association between tinnitus and hyperacusis [50], the self-rating of tinnitus and hearing ability worsened and increased tinnitus modulation [51]. Moreover, in addition to THI and TQ12-I questionnaires, a hyperacusis assessment questionnaire developed by Khalfa and colleagues [52] was included in the methods employed.

Background noise is a well-known condition producing audibility and recognition challenges in hearing-impaired patients, as evidenced by a plethora of studies (e.g., [20,22,29,53,54,55]), but also in normal hearing persons according to the different levels of degradation/difficulty of the auditory condition [3,4,24,28,56,57]. Specifically, the word-in-noise recognition task constitutes a suitable protocol for investigating the listening effort since it elicits the effort necessary to discriminate the speech from the background noise [58]. 

The main aims of the present pilot study therefore were: The investigation of listening effort neurophysiological indices in normal hearing tinnitus patients in comparison to normal hearing controls during a continuous speech stimulus.The assessment of the influence of the different background noise levels on listening effort indices and the difficulty and pleasantness perception of the stimulus by tinnitus participants in comparison to healthy controls.

## 2. Materials and Methods

### 2.1. Participants

In the present pilot study were enrolled 19 participants: 12 chronic tinnitus (TIN) patients (7F, 5M; mean age ± SD: 47.416 ± 12.770) and 7 healthy control (CTRL) participants (4F, 3M; mean age ± SD: 46.314 ± 16.331). Inclusion criteria for all participants were: normal hearing, assessed through pure tone audiometry testing with a pure tone average (PTA) ranging from 125 to 8 kHz up to 20 dB, and absence of major pathologies (e.g., certified psychiatric or neurological pathologies) or anatomo-functional alterations that could affect the study and the absence of psychoactive drugs assumption. The inclusion criterium for the TIN group was the perception of the primary tinnitus symptom (unilaterally and/or bilaterally) for at least 3 months.

### 2.2. Self-Report Audiological Questionnaires

The THI [44,45] is a 25-item instrument developed to quantify the functional and psychosocial consequences of tinnitus and its impact on everyday life, providing supplemental information to the conventional psychoacoustic assessment (e.g., pitch and loudness matching, minimum masking levels, residual inhibition) of tinnitus impairment. In addition to the total score, it is possible to obtain values for three subscales: emotional (E), functional (F), and catastrophic (C). According to the total THI score, tinnitus severity is divided into five categories as follows: no handicap (0–16 points), ‘mild’ (18–36), ‘moderate’ (38–56), ‘severe’ (58–76) or ‘catastrophic’ tinnitus handicap (78–100 points) [41,44]. THI has been widely recommended as a research tool for rating tinnitus severity [59], including EEG evaluation [60].

The Hyperacusis Questionnaire [52] is an internationally validated test (see [61] for Italian validation) and makes it possible to investigate clinical hyperacusis, i.e., unusual tolerance to ordinary environmental sounds [62], while hearing thresholds are often quite normal.

The Tinnitus Questionnaire 12-item short form (TQ12-I) [46] is the short form of the original 20-item test [63] that allows the assessment of tinnitus-related distress. The grade of tinnitus distress according to TQ12-I is categorized as follows: no clinically relevant tinnitus distress (1–7 points); moderately distressed (8–12 points); severely distressed (13–18 points); most severely distressed (>19 points).

### 2.3. Experimental Protocol

The experimental protocol consisted of listening to an audiobook (a short story) previously used in neuroscientific experiments [64] during the simultaneous acquisition of both electroencephalographic and autonomic data. The audio track used in the present study “Storia di Gianna e delle sue chiavi” [https://www.progettobabele.it/AUDIOFILES/ascolta.php?ID=841 (accessed on 23 January 2023)] was taken from the database “Progetto Babele Rivista Letteraria” (http://www.progettobabele.it (accessed on 23 January 2023)). The total duration of the stimulation was 11 min 39 s in 3 signal-to-noise ratio (SNR) randomized conditions: +5; +10; 0 with an average duration of 1 min 31 s and in a quiet condition at the beginning and end of stimulation (average duration 2 min 30 s each). The stimulus was transmitted by two audio speakers placed at 45 degrees left/right, at face level 1 m in front of the participant, as in previous clinical studies in auditory neuroscience [65]. The total auditory stimulation was set at 65 dB [66]. Before the start of auditory stimulation, participants were shown a blank screen for 3 s (pre-stimulus phase). The processing of the audio track in the different SNR conditions was carried out using Audacity software (https://www.audacityteam.org/download/ (accessed on 30 January 2023)), the noise used was the “babble noise” [67] already used in the construction of experimental protocols in auditory neuroscience in normal hearing and hearing-impaired samples [21,22,29,31,55,68]. While listening, the participant was asked to indicate at regular intervals of 90 s, for a total of 7 times, corresponding to the 7 Quiet and SNR conditions (Figure 1), on two separate VAS already used for tinnitus patients [69] with a score from 0 to 100, the subjective perception respectively of perceived pleasantness and difficulty during listening (self-reported data). At the end of the auditory stimulation, a 28-item multiple-choice questionnaire on the story’s content collected the participant’s behavioral responses. A Lenovo PC (monitor resolution 1024 × 768) displayed and controlled the audiobook presentation and collected participants’ responses through the software package E-Prime (Psychology Software Tools, Pittsburgh, PA, USA, Version 3.0). 

### 2.4. EEG Signal Acquisition and Processing

A Mindtooth Touch EEG standard EEG headset with water electrodes (saltwater sponge and passive Ag/AgCl electrodes) (https://www.mindtooth.com/ (accessed on 20 January 2023)) already used for the assessment of psychophysiological variables such as stress in cognitive neuroscience protocols [70] was used for EEG data acquisition. The electrodes used to investigate the scalp area of interest for the study objective corresponded to AFz, AF3, AF4, AF7, AF8, Pz, P3, and P4 of the International 10-10 System defined in [71].

The EEG signal was firstly band-pass filtered with a fifth-order Butterworth filter at 2–30 Hz intervals. The blink artifacts were detected by means of the Reblinca method [72].

The reconstructed EEG signal was then segmented into 1-s-long epochs with 0.5 s of overlap in order to avoid any “boundary effect”, and three additional criteria for detecting artifacts based on the signals’ amplitude and trend [73,74] were applied in order to remove those portions of data still affected by artifacts that had not been corrected before [74].

The Global Field Power (GFP) was calculated for the EEG frequency band of interest, Alpha (8–13 Hz), from the artifact-free EEG. This band was defined accordingly with the Individual Alpha Frequency (IAF) value [75], [IAF-4, IAF+2], estimated specifically from each subject through one minute of eyes closed, which was recorded before starting the experiment.

### 2.5. Autonomic Activity Signal Acquisition and Processing

Electrodermal activity (EDA) was recorded with a sampling rate of 64 Hz through a Shimmer 3 GSR+ (Shimmer Sensing, Dublin, Ireland) system applied to the non-dominant hand of the subject. The constant voltage method (0.5 V) was employed for the acquisition of the EDA. The electrodes were placed on the palmar side of the middle phalanges of the second and third fingers, on the non-dominant hand of the participants, according to published procedures [76]. Employing the LEDAlab software [77], the tonic component of the skin conductance (Skin Conductance Level, SCL) was estimated. The SCL corresponds to the slow-changing component of the EDA signal, consistently related to arousal and stress levels [78].

### 2.6. Statistical Analysis

Given the nature of the pilot study, with a small sample numerosity, non-parametric statistics were employed [79,80], using Friedmann ANOVA for within-group analysis, that is, the testing of the effect of the variable audibility condition (4 levels: Quiet, SNR10, SNR5, SNR0) for the separate comparison within each group (TIN and CTRL). Mann–Whitney U-test was instead employed for between-group comparisons, testing the effect of the variable group (2 levels: TIN, CTRL) on the various indices, assessing the eventual statistically significant difference between the TIN and the CTRL group. Moreover, Spearman Rank Order correlation was employed for testing correlations between neurophysiological values, questionnaire scores, and ratings. For the pre-stimulus phase, the mean of the neurophysiological activity estimated during the 3 s preceding the listening phase was employed. For the Quiet condition, the average value of the four Quiet conditions phases employed in the experimental protocol was calculated (Figure 1). For the statistical analysis, Statistica software (StatSoft, Hamburg, Germany) was employed. All statistical analysis *p*-values ≤ 0.05 were considered statistically significant.

## 3. Results

### 3.1. Behavioural Results

Concerning the Mann–Whitney U-test for comparison between the number of correct responses between groups (TIN and CTRL) for each of the experimental conditions (Quiet, SNR10, SNR5, SNR0), no statistically significant difference was found. Also, the analysis within each group showed no differences among the experimental conditions. 

Concerning the rating scores of the perceived pleasantness and difficulty, there was not any difference between groups for each experimental condition, while there was a within-group difference among the experimental conditions for each group (TIN pleasantness ANOVA Chi Sqr = 9.807, *p* = 0.020; TIN difficulty ANOVA Chi Sqr = 18.310, *p* < 0.001; CTRL pleasantness ANOVA Chi Sqr = 17.609, *p* < 0.001); CTRL difficulty (ANOVA Chi Sqr = 10.454, *p* = 0.015) (Figure 2).

Concerning the THI and TQ12-I questionnaires, average scores were 20.083 (moderate severity) and 8.083 (moderately distressed).

### 3.2. EEG Results

#### 3.2.1. Parietal Alpha

During the pre-stimulus phase, parietal alpha levels did not differ between groups, but in the TIN group, a moderate correlation between parietal alpha levels and the number of total correct responses was found (r = 0.582, *p* = 0.047) and a strong correlation with the number of the correct responses for the Quiet condition (r = 0.671, *p* = 0.017). Differently, in the CTRL group, pre-stimulus parietal alpha levels strongly correlated with the hardest auditory condition (SNR0) correct responses (r = 0.805, *p* = 0.029). Moreover, in the TIN group, there were negative correlations between pre-stimulus parietal alpha levels and the perceived difficulty ratings for all the conditions (Quiet r = −0.735, *p* = 0.006; SNR10 r = −0.879, *p* < 0.001; SNR5 r = −0.762, *p* = 0.004) except for the hardest one (SNR0 r = −0.315, *p* = 0.319). In addition, in the TIN group, pre-stimulus parietal alpha levels were also negatively strongly correlated (r = −0.720, *p* = 0.008) with hyperacusis scores as indexed by the Khalfa questionnaire [52] (Figure 3).

Concerning the estimated levels of parietal alpha during listening to the audiobook under the different auditory conditions, there were no statistically significant differences between groups. The Friedman ANOVA analysis showed no effect of the variable condition on the parietal alpha levels in the TIN group, but for the CTRL group, it was found that the effect of the variable condition just missed statistical significance, in particular with decreasing parietal alpha from the easiest (Quiet) to the most challenging condition (SNR0) (ANOVA Chi Sqr = 0.763, *p* = 0.054).

However, only for the TIN group, there was a negative correlation between parietal alpha levels and EDA values in all the conditions (Quiet r = −0.727, *p* = 0.007; SNR5 r = −0.615, *p* = 0.033; SNR0 r = −0.706, *p* = 0.010) except for SNR10 (r = 0.419, *p* = 0.174) (Figure 4). Moreover, it was found a correlation in the TIN group between parietal alpha levels and TQ12-I scores in all the conditions: Quiet r = 0.749, *p* = 0.005, SNR10 r = 0.749, *p* = 0.005, SNR5 r = 0.815, *p* = 0.001, SNR0 r = 0.626, *p* = 0.029 (Figure 4).

#### 3.2.2. Frontal Alpha

Analogously to parietal alpha, we investigated the cortical activity preceding the expected stimulus (audiobook) for frontal alpha. There was no statistically significant difference between the TIN and the CTRL group for the pre-stimulus frontal alpha levels. The TIN group found a correlation between pre-stimulus frontal alpha levels and THI scores (r = 0.599, *p* = 0.040), with higher pre-stimulus frontal alpha levels linked to higher THI scores (Figure 5).

Concerning the frontal alpha levels during the different conditions, there was a general trend of increased alpha activity in the TIN in comparison to the CTRL group (Mann–Whitney U-test: Quiet *p* = 0.031; SNR10 *p* = 0.083; SNR5 *p* = 0.038; SNR0 *p* = 0.057) (Figure 6). Finally, no effect of the variable condition in the within-group analysis was observed for both the TIN and the CTRL group.

### 3.3. EDA Results

No differences existed between groups for the EDA levels in all the conditions, even in the pre-stimulus phase. In the TIN group EDA levels were negatively correlated with TQ12-I in almost all the conditions (Quiet r = −0.590, *p* = 0.043; SNR5 r = −0.626, *p* = 0.029; SNR0 r = −0.843, *p* < 0.001), except for SNR10 (r = 0.105, *p* = 0.744). There were no within-group differences concerning EDA levels.

## 4. Discussion

The absence of statistically significant differences in the behavioral data between the CTRL and TIN groups (*p* > 0.05) could be because, on average, the perception of tinnitus assessed with the THI was moderate, not severe or catastrophic. In addition, the TQ12-I average score was moderately distressed and not severely or most severely distressed. Perceived severity and distress, therefore, failed to impact behavioral performance, in line with the review of the impact of tinnitus severity on behavioral performances in cognitive tasks [81]. We can assume that the disconfirmation due to the clinical condition is not yet reflected in a negative behavioral pattern. However, the results show patterns at the neurophysiological level.

For the tinnitus group, the negative correlation between parietal alpha during the listening and the EDA levels was supported by a previous comparative study showing the same negative correlation [82]. The same authors state the multidimensional nature of listening effort measures, among which measures were scarcely correlated [82], therefore providing complementary but not overlapping measures of listening effort. Also, the negative correlation, found in the tinnitus group, between the perceived Difficulty in almost all the conditions (except the hardest one: SNR0) and the pre-stimulus alpha levels were found by Alhanbali and colleagues through a factorial analysis showing these two measures resulted in being included into the same factor (factor 4) in a factor analysis performed over many listening effort measures [82]. An analog discussion could be made for performance and pre-stimulus alpha levels, resulting in the correlation of the present study in the tinnitus group for the easiest condition (Quiet) correct responses with the overall task correct responses; baseline alpha levels and correct responses have been associated in factor 1 as a result of the factor analysis mentioned above [82]. Moreover, according to Klimesch and colleagues [17], increased baseline alpha activity is an indicator of pre-task cortical engagement that predicts improved task performance. Here it is possible to speculate that the correlation between pre-stimulus alpha and performance in the TIN group, specifically in the simplest condition, may be due to the increased cortical resources required to cope with listening in the “external” quiet listening. Considering the absence of such a correlation in the CTRL group, this could be a prodrome marker of the greater listening effort perceived even in the absence of external noise, given that the “internal” tinnitus noise is still present. Tinnitus can be considered a form of internal noise that may affect speech recognition in both ears through central interference [83].

Concerning the correlation between pre-stimulus frontal alpha levels and THI score, a study investigating the distress network in chronic tinnitus patients showed a positive correlation between at-rest alpha activity over the prefrontal cortex and the THI [60].

Given the negative correlation (Figure 3) between pre-stimulus parietal alpha levels and Khalfa questionnaire scores, hyperacusis seems to negatively influence the cerebral activity aimed at preparing the reactivity to a stimulus to obtain better performances. Such a hypothesis is supported by a previous study showing that normal hearing patients with tinnitus and diagnosed hyperacusis performed worse than the control group in speech recognition in the presence of competitive noise [84]. On the contrary, given the positive correlation (Figure 3) between pre-stimulus parietal alpha levels and the number of total correct responses, parietal alpha activity before an expected stimulus appears to be confirmed to play a preparatory role for performances [21,24].

Furthermore, parietal alpha levels during listening to the auditory stimuli, positively correlated to the TQ12-I (Figure 4), appear to reflect a higher listening effort in tinnitus patients and higher distress related to tinnitus symptoms perception.

The negative correlation between both listening effort (parietal alpha) and tinnitus symptoms perception (TQ12-I scores) with EDA levels could be explained by the less responsive sympathetic nervous system to prepare the body to expend increased energy during the “fight or flight” response [68], due to the pauperization of such energy from tinnitus perception, as deducible from the cascade of negative psychophysical effects caused by tinnitus [85]. Moreover, chronic stress can lead to an “allostatic load” that could be reasonably produced in persons with hearing loss, subjected to increased and sustained cognitive load and stress [10]. The present study advanced the hypothesis of generalizing such phenomena from hearing loss to a condition characterized by normal hearing but an auditory disorder, tinnitus. The hypothesis above is also supported by the lack of habituation retrieved in normal hearing tinnitus patients [86] and the lack of correlation between parietal alpha and EDA values in the control group.

TQ12-I appears to be more sensitive to tinnitus symptoms perception during auditory task execution, while THI seems more sensitive to them regarding a preparatory/baseline phase. This could be explained by the hypothesis that TQ12-I would better reflect the functional-cognitive dimension of tinnitus-related distress. At the same time, the THI would be more informative about general task-independent symptomatology. Finally, as predicted, for (normal hearing) tinnitus patients, the listening effort appears to be more related to tinnitus symptoms perception than environmental audibility conditions, given the lack of statistical differences between quiet and background noise conditions. Given the nature of the pilot study of the present research, a limitation of the study is the limited sample size. Therefore, further investigations are needed to confirm the current results and test additional tinnitus groups characterized by hearing loss and/or different sample stratification.

## 5. Conclusions

The results of the present pilot study led to some preliminary conclusions concerning the above-mentioned main objectives of the research:1.The investigation of listening effort neurophysiological indices in normal hearing tinnitus patients in comparison to normal hearing controls during a continuous speech stimulus.Despite a general lack of difference between the TIN and CTRL groups concerning performances and perceived pleasantness or difficulty, possibly explained by the normal hearing condition shared by both groups, neurophysiological patterns and correlations retrieved in the TIN group support the hypothesis of a relation between listening effort underpinnings and tinnitus symptoms. The relevance of employing continuous speech stimuli presents a step forward in identifying neural patterns mirroring the daily communication conditions experienced by tinnitus patients.2.The assessment of the influence of the different background noise levels on listening effort indices and the difficulty and pleasantness perception of the stimulus by tinnitus participants in comparison to healthy controls.In the present study, the level of SNR appears to not influence the listening effort experienced by the TIN group, probably due to the normal hearing condition chosen for patients, a condition purposefully selected to avoid biases due to the hearing-impaired condition effect on listening effort. Such a result should be further investigated on an enlarged sample.

## Figures and Tables

**Figure 1 brainsci-13-01084-f001:**
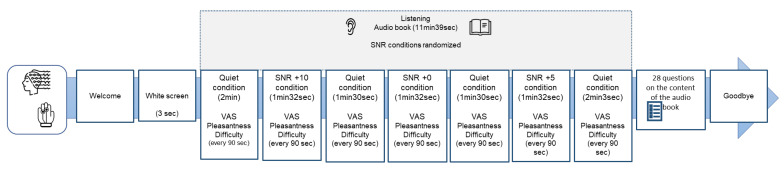
Graphical representation of the experimental protocol employed in the present pilot study. The auditory conditions during the audiobook listening were pseudorandomized among participants, maintaining the sequence Quiet–signal-to-noise ratio (SNR)–Quiet–SNR–Quiet–SNR–Quiet, resulting in 7 conditions, and for correspondence of each of them, 4 questions have been made at the end of the listening phase in order to assess comprehension. Every 90 s, a visual analog scale (VAS) concerning perceived pleasantness and a VAS concerning perceived difficulty were presented to the participant.

**Figure 2 brainsci-13-01084-f002:**
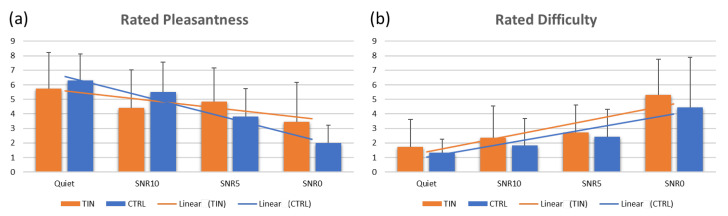
Bar graphs representing the average rating scores for the perceived Pleasantness (**a**) and Difficulty (**b**) given by the tinnitus (TIN) and the control (CTRL) group in correspondence of each auditory signal-to-noise ratio (SNR) condition (Quiet, SNR10, SNR5, SNR0). Error bars stand for standard deviations.

**Figure 3 brainsci-13-01084-f003:**
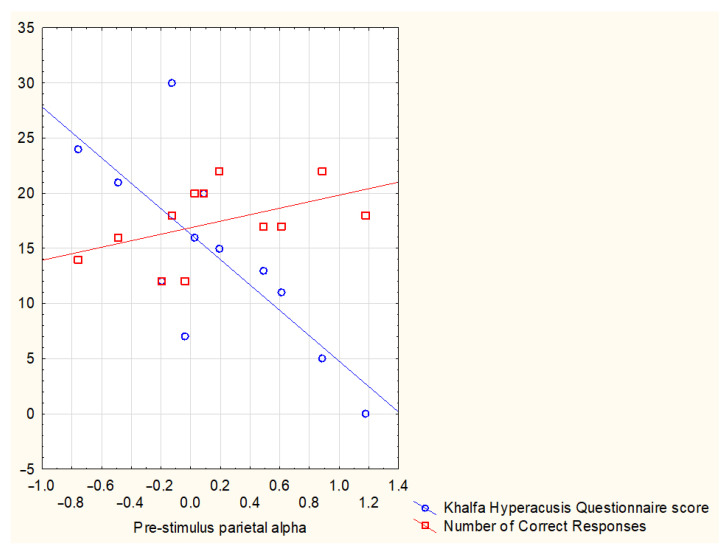
Scatterplot representing the correlation between pre-stimulus parietal alpha levels and, respectively, the number of correct responses for the entire task irrespective of the specific auditory condition (Quiet, SNR10, SNR5, SNR0) and the score reported in the hyperacusis questionnaire by Khalfa and colleagues [52].

**Figure 4 brainsci-13-01084-f004:**
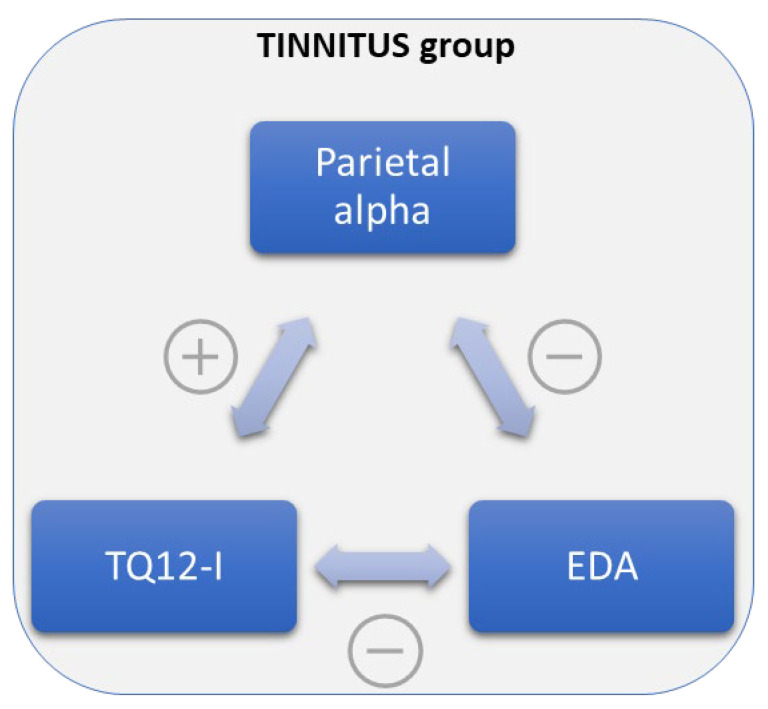
Scheme of the general positive 
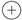
 and negative 
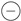
 correlations between Parietal alpha levels, Tinnitus Questionnaire 12-item short form (TQ12-I) scores, and electrodermal activity (EDA) levels in the chronic tinnitus (TIN) group.

**Figure 5 brainsci-13-01084-f005:**
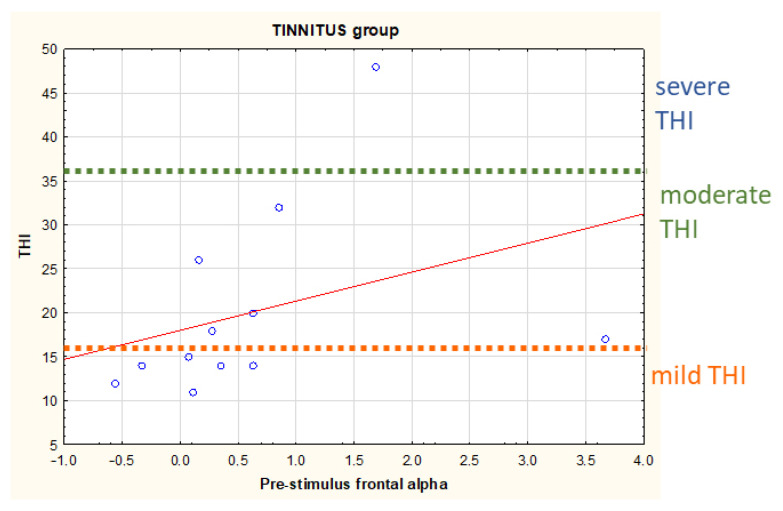
Scatter plot representing the correlation between pre-stimulus frontal alpha levels and the Tinnitus Handicap Inventory (THI) scores in the TIN group. For convenience it was also reported on the right the corresponding threshold for the different THI scores: mild (18–36), moderate (38–56), and severe (58–76 points) reported by the participants to the study; no one reported ‘catastrophic’ tinnitus handicap scores (78–100 points).

**Figure 6 brainsci-13-01084-f006:**
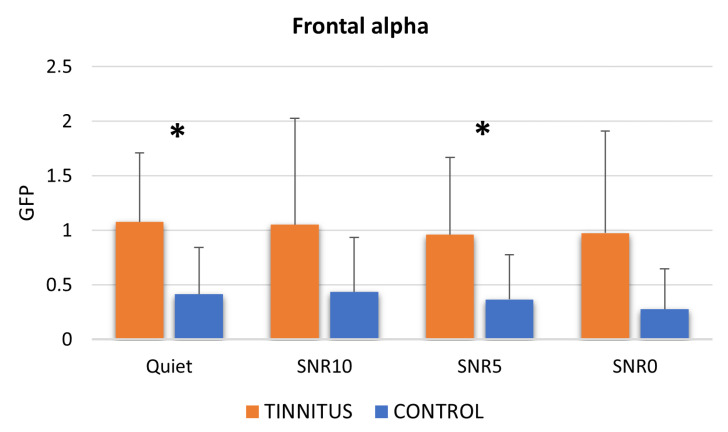
Graph representing the mean frontal alpha values estimated for each group in each condition and the statistically significant differences resulting from the Mann–Whitney U-test between TIN and CTRL groups (Quiet, SNR10, SNR5, SNR0). * stands for *p* < 0.05. Error bars stand for standard errors.

## Data Availability

The raw data supporting the conclusions of this article and the material will be made available by the authors without undue reservation.

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
