# Peer review of "Listening Effort in Tinnitus: A Pilot Study Employing a Light EEG Headset and Skin Conductance Assessment during the Listening to a Continuous Speech Stimulus under Different SNR Conditions"

_brainsci, 2023, doi:10.3390/brainsci13071084_

Round 1
Reviewer 1 Report
Comments and Suggestions for Authors
In this manuscript the authors designed a very interesting project on the occurrence of increased listening effort as a hot topic in hearing research in tinnitus patients during listening task investigating some indices through electroencephalographic and skin conduction. The language seems correct on the whole of the paper and the cited references relevant to the research. The results are similar to a previous comparative study concerning the negative correlation between parietal alpha during the listening and the electrodermal activity levels for the tinnitus group.
Given the small sample size of this pilot study, it was employed non-parametric statistics for within group analysis, between group comparisons, correlations between neurophysiological values, questionnaire score and rating. An increased sample might confirm more pronounced the present results testing further tinnitus groups characterized by hearing loss and also different stratification of the sample.
Comments on the Quality of English LanguageThe language is in general correct.
Author Response
We deeply thank the Reviewer for the interested toward our study!
The sample size was limited due to the pilot nature of the study, however we applied non-parametric statistical tests, that given their more conservative level than parametric test are recommended by some statisticians in order to prevent type I and II errors (e.g. Siegel, S. (1957). Nonparametric statistics. The American Statistician, 11(3), 13-19.; Weber, M., & Sawilowsky, S. (2009). Comparative Power Of The Independent t, Permutation t, and WilcoxonTests. Journal of Modern Applied Statistical Methods, 8(1), 3.).
Reviewer 2 Report
Comments and Suggestions for Authors
First, I would like to thank you for reading and reviewing this manuscript. The topic and results of the present investigation are interesting and add to knowledge; however, many issues must be corrected before it can be published. Regarding this, see my specific recommendations in the following.
Keywords. Instead of alpha, alpha values would be more accurate.
In the introduction, please, explain all abbreviations in brackets (must be included in the other parts of the text too). Always explain the abbreviations by their first appearance, and then, consequently use only the abbreviation through the text.
Lines 43-47. I recommend removing ‘e.g.’ from this sentence.
Lines 60-61. This sentence regarding the areas of alpha increasing should be placed after the sentence introducing the alpha activity increase, i.e., lines 56-58. After that, the authors can introduce the alpha activity decreasing.
Generally, in the introduction, I miss some basic information on tinnitus. E.g., the definition of tinnitus, subjective/objective types, etc., must be explained. This can be included with the following reference article:
[Molnár A, Mavrogeni P, Tamás L, Maihoub S. Correlation Between Tinnitus Handicap and Depression and Anxiety Scores. Ear Nose Throat J. 2022 Nov 8:1455613221139211. doi: 10.1177/01455613221139211.]
Lines 103-105. Instead of st. dev., SD is a better abbreviation. Furthermore, I always recommend including numbers to two places of decimals.
Lines 107-108. I miss some information on the inclusion criteria for tinnitus patients. E.g., were only primary tinnitus patients included, and if yes, how were the secondary cases excluded? Patients with one- or both-sided symptoms were investigated? I find it essential to explain the inclusion/exclusion criteria for both groups in more detail.
Line 109. I do not find this is the best title for this paragraph. Audiological measurements were not carried out based on the methods described in this section; self-reported questionnaires were used. These are essential in the diagnosis and follow-up of patients with tinnitus. A subtitle like ‘Self-reported questionnaires’ would be more accurate.
Lines 110-115. I would explain here the subscales and rating of the THI. A useful reference article for this: [Mavrogeni P, Maihoub S, Tamás L, Molnár A. Tinnitus characteristics and associated variables on Tinnitus Handicap Inventory among a Hungarian population. J Otol. 2022 Jul;17(3):136-139. doi: 10.1016/j.joto.2022.04.003.]
Was the Italian version of the THI validated? Please state.
This paragraph suggests to me that under audiological measurements, the authors mean the use of the present questionnaire. However, the use of at least pure-tone audiometry and tinnitus pitch matching is always essential in the case of diagnosis of tinnitus. If hearing testing was performed, then include its methodical details. If not, then include this, but in this case, this must be stated as a significant limitation of the present study. In the previous parts of the manuscript, it was stated that only participants with normal hearing were included. Therefore, please clarify whether this was based on the participants’ subjective reports or on audiological testing.
Line 129. Explain ‘SNR’ abbreviation at its first appearance.
Figure 2. Include the abbreviations of the figure in the figure caption.
Statistical analysis
Regarding the statistical analysis, some details are missing. First of all, please clarify which test was used for what reason. Friedman ANOVA test is a non-parametric test, the use of which suggests a not normal distribution of the data. It can be used to compare more than two groups of variables (similarly to the ANOVA test, which is used for the same in the case of a normal data distribution). The Mann-Whitney U test is non-parametric test as well, but is used for the comparison of two groups of variables. From my point of view, the current presentation of the used statistic methods might be misleading. Please include the name and manufacturer of the statistical software. Furthermore, the use of non-parametric tests suggests a not normal distribution of the data, but it is not stated how this was analysed. In addition, in the results, the Chi-square test is also mentioned; however, it cannot be found in the methods. In addition, please state what significance level was applied.
Figure 3. Indicate that in the bar charts, the mean values are presented. Furthermore, please use the English version of ‘linear’ (instead of ‘lineare’). Include the abbreviations in Figure caption in this case as well.
Lines 195-198. Based on which statistical method were these non-significant differences observed?
Lines 201-203. The statistical analysis regarding the within-group difference needs to be clarified. The Chi-square test is used to compare categorical variables within four or more groups. Please clarify this. Furthermore, when Figure 3. is referenced in the text, the explanation is not in accordance with the results in the Figure, i.e., the mean and SD values of each group).
Line 212. Was this correlation statistically significant? If yes, include the p-value. Furthermore, I recommend mentioning that this is a moderate correlation according to the interpretation of Spearman’s rho. The same is necessary in the case of the other correlation test results; please correct the consequently.
Figure 4. Calculate the R2 values of the linear correlations.
Line 226. ‘Within group’ analysis is, once again, not the best terminology. Furthermore, the other parts of this sentence are not easy to follow.
Lines 226-229. I recommend rephrasing this sentence.
Lines 230-234. The p-values regarding the correlation analyses are missing.
Figure 5. Explain the abbreviations in the Figure caption.
Figure 6. In the figure caption, include the ranges of the THI categories. This also should be included in the methods, as previously mentioned. R2-value would be interesting here too.
Line 258. Indicate that significant differences were detected based on the Mann-Whitney U test.
At the end of the discussion, the limitations of the present investigation must be included (e.g., the low number of participants, the lack of using audiometry if it was not used, etc.). Although this was a pilot study, these must be included.
In the conclusion, the authors should try to emphasise the practical relevancies of this study.
Lines 346-349. Include the number of ethical approval.
Generally, English grammatical mistakes and typos must be corrected throughout the text.
I look forward to receiving the revised version of the manuscript.
Comments on the Quality of English LanguageGenerally, English grammatical mistakes and typos must be corrected throughout the text.
Author Response
Dear Reviewer,
many thanks for your precious work, please find the attached file.
Best regards,
Giulia Cartocci

Round 2
Reviewer 1 Report
Comments and Suggestions for Authors
Published in revised version
Reviewer 2 Report
Comments and Suggestions for Authors
Thank you for the revised version of the manuscript; the quality of the manuscript has been significantly improved.
Comments on the Quality of English LanguageMinor editing of English is required.